# Taxonomic Identification and Molecular DNA Barcoding of Collected Wild-Growing Orchids Used Traditionally for Salep Production

**DOI:** 10.3390/plants12173038

**Published:** 2023-08-24

**Authors:** Aphrodite Tsaballa, George Kelesidis, Nikos Krigas, Virginia Sarropoulou, Panagiotis Bagatzounis, Katerina Grigoriadou

**Affiliations:** 1Hellenic Agricultural Organization Demeter (ELGO-DIMITRA), Institute of Plant Breeding and Genetic Resources, Thermi, 57001 Thessaloniki, Greece; atsampalla@elgo.gr (A.T.); gkthgk@gmail.com (G.K.); nkrigas@elgo.gr (N.K.); vsarrop@gmail.com (V.S.); 2‘Spices Bagatzounis’ Company: El Greco, Natural Herbs & Teas, Vatero, 50100 Kozani, Greece; panos@bagatzounis.com

**Keywords:** *Dactylorhiza*, *Orchis*, *Anacamptis*, *Himantoglossum*, Orchidaceae, non-timber forest products (NTFP), wild products, salep, illegal trade, conservation value

## Abstract

Molecular DNA barcoding combined with botanical taxonomy can be used for the identification and conservation of collected Greek orchids used for salep production as well as in the regulation of fair salep trade. A modified CTAB protocol was used for DNA extraction, amplification of barcoding regions (*ITS*, *matK*, *rbcL*, *trnH-psbA*), and sequencing. Sequencing data were assembled using Bioedit software, and the BLAST algorithm was used on the NCBI database for species identification at the genus level. Molecular barcoding data based on genetic similarity identification was in full coherence with taxonomic classification based on morphological data. The combination of *ITS* and *matK* exhibited a greater capacity to identify a species among the Greek salep samples. Out of the 53 samples examined, 52.9% were classified as *Dactylorhiza* spp. and 33.3% as *Anacamptis* spp., whereas only 6 samples were identified as *Orchis* spp. (11.8%). Given that a superior-quality salep beverage comes from tubers of the latter, the number of samples classified as such in northwestern Greece is unexpectedly low. A database of 53 original reference sequences from wild-growing samples of Greek origin was generated, providing a valuable resource for the identification of other salep samples from different regions. The DNA barcoding results unveiled that salep samples from northwestern Greece are related to nine members of four different genera of Orchidaceae. All species are nationally protected and covered by the CITES convention, while many of these orchids are included in the EU Directive 92/43/EEC appendix as “Other Important Species”. Thus, expedited coordinated management actions are needed to ensure their survival in the future.

## 1. Introduction

Apart from including attractive plants, the family Orchidaceae, comprising about 736 genera and 28,000 species worldwide [1], is rather extraordinary since its members have been included in Appendices of the Convention on International Trade in Endangered Species, namely, CITES (https://cites.org/eng, accessed on 1 May 2023) [2]. To date, many orchid species are highly represented in either conventional or electronic commerce over the internet, thus being traded legally or not for their high ornamental value or as a source of components for cosmeceuticals and herbal medicines and as food [3,4]. Consequently, numerous orchid species are long-known in the folk tradition of many nations around the world [5,6,7]. 

The commercial product commonly called ‘salep’ (‘salepi’ in Greek) is the most famous folk preparation comprising orchids [8]. For thousands of years, terrestrial orchids in the Eastern Mediterranean and the Balkans have been harvested to ground their tubers to produce the famous salep powder [7,9,10,11,12,13,14,15,16,17,18]. This valuable substance is traditionally served as a hot drink, which is consumed especially during winter, or it is used as a basic ingredient for the derivate ice cream called ‘salep dondurma’ in Turkey or ‘Kaimaki’ in Greece [9,10,11,14,19]. Although this preparation originated from the Eastern Mediterranean and the Balkans, it became famous across Europe during the Renaissance period following the trend after the publication of *Gerard’s Herbal* in 1633 [7]. To date, it has been reported that thousands of orchid bulbs are widely harvested every year for salep in different regions [19,20,21]. In the past, orchid tubers were commonly collected in northern Greece for the preparation of salep beverages and gelatin for porridge, representing a working-class staple [22]; however, overharvesting still occurs. This harvesting remains common in Turkey, the Balkans, and Iran [7,10,15,19]. In Greece, all orchid species are protected at the national level by the Greek Presidential Decree 67/1981 and are covered by the CITES convention. Their collection from the wild is forbidden and banned and is thus considered illegal [23]. 

Currently, the most popular form of salep is a hot beverage that is consumed to soothe the throat and ease stomach aches and intestinal cramps and as a remedy for colds and coughing [14]. Several orchid species are commonly collected from northern Greece for salep production such as *Anacamptis coriophora* (L.) R.M. Bateman, Pridgeon & M.W. Chase, *Anacamptis morio* (L.) R.M. Bateman, Pridgeon & M.W. Chase, *Anacamptis papilionacea* (L.) R.M. Bateman, Pridgeon & M.W. Chase, *Orchis anthropophora* (L.) All., and *Orchis italica* Poir. [22], or *Anacamptis pyramidalis* (L.) Rich., *Dactylorhiza sambucina* (L.) Soó, *Dactylorhiza saccifera* (Brongn.) Soó, *Orchis militaris* L., *Orchis provincialis* Balb. ex Lam. & DC., and *Orchis simia* Lam. [14]. One way or another, the starch contained in these salep orchids may interact with arabin and tragacanthin substances when diluted in water or milk, creating a thick and sticky liquid due to the contained substance vassarin; the latter seems to have a soothing effect against coughs and asthma [5]. Among different orchid species, the best salep is considered to be the one derived from *Orchis mascula* [14]. Apart from strengthening and stimulating the body and soothing stomach aches, *Orchis mascula* salep has presumed lung anticancer activity, strong antioxidant activity, and beneficial effects on hyperlipidemia and hypertension [24]. Undoubtedly, salep, or salepi in Greek, is an ideal high-energy drink due to the additional spices it contains (e.g., cinnamon) and its high content of carbohydrates and valuable elements such as phosphorus and calcium [24].

Given the ethnic importance of salep and the concomitant concerns for orchid conservation [7,10,15,19], several investigations have already focused on this nutritious and valuable yet controversial product with largely unproven health or aphrodisiac effects [8]. Previous review studies [8] report as many as 46 tuberous and 1 rhizomatous orchid species sourced from the wild for salep in the European context. However, other studies from Turkey have reported that up to 90 different taxa (species and subspecies) may be harvested for this aim [10], and 38 species from 7 genera were reported to be harvested in Iran [12,13]. In one way or another, several members of the genera *Orchis*, *Anacamptis*, and *Ophrys* and also *Dactylorhiza*, *Himantoglossum*, *Neotinea*, *Platanthera,* and *Serapias* are reported as common among these studies [8].

The above-mentioned studies coupled with the complex taxonomy of Orchidaceae [25,26,27,28] imply that advanced taxonomic identification is needed to elucidate salep composition from different regions. The use of DNA sequences as “barcodes” constitutes a fast, dependable, low-cost, and straightforward solution for the identification of species [29,30]. In land plants, short regions of nuclear DNA, such as the internal transcribed spacer (*ITS*), and chloroplast DNA, such as *rbcL*, *maturase K* (*matK*), *psbA*-*trnH*, and the transfer ribonucleic acid leucine (*trnL*), are broadly used as markers for the molecular identification of plant species [31]. However, in the complex and large family of Orchidaceae, more regions of chloroplastic DNA have been additionally used for the classification of species such as *psaB*, *trnL-F*, and nuclear Xdh [25,26,27]. In general, the simultaneous utilization of coding and non-coding regions is necessary for the successful identification of species in the Orchidaceae family [26,32,33,34]. Previous studies have tested several barcoding regions and suggested that the combination of *ITS* and *matK* markers can successfully identify members of the large genus *Dendrobium* in the family Orchidaceae [28]. Other studies have used *nrITS*, *trnL-F*, and *matK* for barcoding DNA extracted from 150 collected tubers of Iranian orchids used to produce salep [35], revealing that most Iranian tubers belonged to species in the genera *Orchis*, *Anacamptis*, and *Dactylorhiza*. Specifically, *nrITS* and the *trnL-F* spacer proved to be easier to amplify and sequence than the *matK*, providing a better-discriminating ability that eventually led to the recognition of the species that are most threatened by overharvesting in different regions of Iran [35]. Thus the latter has shown how DNA barcoding may aid the onset of conservation strategies. 

Genomics technologies and approaches are of vital importance to maintain and conserve biodiversity as they can provide reliable results, even if DNA is too degraded and difficult to sequence using next-generation tools [36]. However, the analytical methods and specific sequences for DNA barcodes are still limited for large families such as Orchidaceae [37]. With the development of molecular biology and bioinformatics, a more improved integrated analytic method for DNA barcoding can be established to identify and distinguish different species [38].

The scope of this study was first to check whether widely used DNA barcoding markers already used in other Orchidaceae studies can be used for the identification of collected wild-growing orchids used for salep by the Sarakatsani ethnic Greek population subgroup in northwestern Greece [14]. The second goal was to assign the collected orchids to specific genera or species using the combination of molecular DNA barcoding and botanical taxonomic identification. The final objective was to set up a specific, easy, and straightforward DNA barcoding protocol that can aid the identification and conservation of Greek orchids, with the aim to incorporate molecular genomic techniques such as DNA barcoding and other taxonomic methods in the regulation of fair salep trade. 

## 2. Results and Discussion

DNA was extracted from 19 fresh and 32 dried plant tissue samples of wild-growing Greek orchids depending on the availability. The barcoding regions *ITS* and *matK* were successfully amplified, sequenced, and used for the identification of almost all samples, while three barcoding regions (*ITS*, *matK* and *trnH-psbA*, or *rbcL*) were successfully used for only four samples. Although in most samples the *trnH-psbA* region was amplified and sequenced, the results were in some cases divergent from the results of the other two barcoding regions that were totally in agreement. Divergent results are accompanied by low BLAST similarities with our samples. A lack of information or taxonomic misclassifications in public DNA databases could lead to divergent results and misidentifications. In only a few samples, *trnH-psbA* was used instead of one of the other two regions for molecular identification. No dependence of the sequencing success rate on the origin of the sample (fresh or dried tissue) was observed. The combination of *ITS* and *matK* demonstrated a greater capacity to identify a species among the Greek salep samples, in agreement with previous studies concerning the family Orchidaceae [32]. Conflicting results were reported before in the literature between DNA barcoding markers within the family Orchidaceae [27]. The taxonomic identification of samples was not possible in only a few cases due to inappropriate (out-flowered) specimens collected from the wild (Table 1, Appendix A). While the taxonomic identification identified some specimens as *Orchis* sp. due to out-flowering appearance (Table 1, Appendix A), these identifications were further identified with molecular barcoding data as *O. pallens* L. (GR-1-BBGK-21,239; GR-1-BBGK-21,240; GR-1-BBGK-21,241; GR-1-BBGK-21,242; GR-1-BBGK-21,244) or *O. quadripunctata* Cirillo ex Ten. (GR-1-BBGK-21,243). In some other cases, the molecular barcoding data suggested the identification as *Dactylorhiza maculata* (L.) Soó or *Dactylorhiza sambucina* (L.) Soó for the specimen GR-1-BBGK-18,6097-26, and *D. maculata* for the specimens GR-1-BBGK-19,6097-27 and GR-1-BBGK-16,6097-29 (Table 1 and Appendix A). In another six cases, both the taxonomic identification and the molecular barcoding data identified the specimens only at the genus level as *Dactylorhiza* sp. (GR-1-BBGK-18,6097-1; GR-1-BBGK-18,6097-9; GR-1-BBGK-18,6097-20; GR-1-BBGK-18,6097-30; GR-1-BBGK-18,6097-32), and the specimen GR-1-BBGK-18,6097-3 as *Anacamptis* sp. (Table 1 and Appendix A). Except for one sample (19,402) identified as *Anacamptis pyramidalis* (L.) Rich., all other samples (*n* = 16) were found to be *Anacamptis morio* (L.) R.M. Bateman, Pridgeon & M.W. Chase with molecular barcoding data; the latter belong to subsp. *caucasica* (K.Koch) H. Kretzschmar, Eccarius & H. Dietr. as determined using the taxonomic identification of the collected samples (Table 1 and Appendix A). 

The taxonomically identified *Dactylorhiza majalis* (Rchb.) P.F. Hunt & Summerh. subsp. *pythagorae* (Gölz & H.R. Reinhard) H.A. Pedersen, P.J. Cribb & Rolf Kühn (synonym *D. kalopissii* E. Nelson subsp. *pythagorae* (Gölz & H.R. Reinhard) Kreutz; sample GR-1-BBGK-21,146) and *Dactylorhiza sambucina* (samples GR-1-BBGK-22,59 and GR-1-BBGK-18,6097-7) were only identified as *Dactylorhiza* sp. with molecular barcoding data due to contradictive matching results in the NCBI database. The taxonomically identified sample GR-1-BBGK-18,6097-24 as *D. sambucina* was matched as either *D. sambucina* or *D. incarnata* (L.) Soó with molecular barcoding data; however, it is highly possible that this sample may be *D. sambucina* due to *ITS* barcoding accuracy (Table 1, Appendix A). The taxonomically identified *Dactylorhiza maculata* (L.) Soó subsp. *saccifera* (Brongn.) Diklic (synonym of *D. saccifera* (Brongn.) Soó subsp. *Saccifera*) was only identified at the species level with molecular barcoding data (*n* = 6 samples); a possible molecular match with *D. maculata* subsp. *fuchsii* (Druce) Soó for the specimen GR-1-BBGK-21,119 (Table 1, Appendix A) was disregarded since this subspecies is absent from Greece (https://portal.cybertaxonomy.org/flora-greece/intro, accessed on 1 May 2023) [39]. In another case, the possible matching of samples GR-1-BBGK-18,6097-2, GR-1-BBGK-18,6097-18, and GR-1-BBGK-18,6097-29 with either *D. sambucina* or *Dactylorhiza viridis* (L.) R.M.Bateman, Pridgeon & M.W.Chase (synonym *Coeloglossum viride* (L.) Hartm.) was rejected (Table 1, Appendix A) due to the absence of key characters in the examined samples such as yellow-green or tinged purple flowers. Finally, the taxonomically identified sample GR-1-BBGK-22,61 as *Himantoglossum calcaratum* (Beck) Schltr. subsp. *rumelicum* (H.Baumann & R.Lorenz) Niketic & Djordjevic (synonym *Himantoglossum jankae* Somlyay, Kreutz & Óvári) was only identified at the genus level with molecular barcoding data (Table 1, Appendix A) due to possible matching with *H. caprinum* (M.Bieb.) Spreng. (synonym *H. affine* (Boiss.) Schltr.); however, the latter is not reported as currently present in the Greek territory (https://portal.cybertaxonomy.org/flora-greece/intro, accessed on 1 May 2023) [39]. In general, the molecular barcoding data based on genetic similarity identification was in full agreement with the taxonomic classification based on morphological data (Table 1, Appendix A). 

Using the BLAST algorithm on the NCBI database, we were able to identify all the Greek orchid samples collected at the genus level. Some samples were identified at the species level, but none were identified at the subspecies level. In general, from the 53 samples examined, half of them were classified as species of the genus *Dactylorhiza* (52.9%). Seventeen samples were classified as *Anacamptis* (33.3%) (Figure 1). In many cases, our sequences were found to be highly similar to the sequence KU931620 deposited in GenBank (Orchidaceae member AG-2017 voucher T20); according to Ghorbani et al. [35], the latter sequence comes from a tuber in Tehran, Iran, that was characterized as *Anacamptis morio*. Nonetheless, all Greek samples examined herein were taxonomically identified as *Anacamptis morio* subsp. *caucasica*. Salep samples from Iran have been reported to include mainly different members of the genus *Orchis* [35]. Surprisingly, in this investigation, only six samples were identified as *Orchis* (11.8%) (Figure 1).

Given that superior-quality salep beverages come from tubers that belong to *Orchis* spp., the number of samples classified as such in the sample set from northwestern Greece is unexpectedly low. According to the BLAST results, the Greek samples probably belong to *Orchis pallens* or *Orchis quadripunctata*. A closer inspection of the *ITS* barcoding region nucleotide alignment analysis indicates that the Greek samples are highly similar to *O. pallens*, and thus are different from other *Orchis* spp. (indicative arrows in Figure 2). 

However, according to the phylogenetic analysis of the *matK* barcoding region, the same Greek samples were placed close to *O. quadripunctata* and *O. mascula* (Figure 3). Still, there is a lack of *O. pallens matK* regions deposited in NCBI that could facilitate an accurate molecular identification. The *trnH-psbA* barcoding region was of no use for these samples as it produced completely contrasting results with the other two barcoding regions (Appendix A). All these samples were collected from one specific geographical area, and their barcoding sequences were almost identical, thus proving their close taxonomic proximity. 

Collectively, this study generated a database of 53 original reference sequences from wild-growing samples of Greek origin, thus providing a valuable resource for the identification of other salep samples from different regions. The results of DNA barcoding using applied markers revealed that salep samples from northwestern Greece represent nine members of four different genera of Orhidaceae (Figure 1). Other review studies [8] reported as many as 46 tuberous and 1 rhizomatous orchid species used for salep in the European context, up to 90 different taxa in Turkey [10], and up to 38 species from 7 genera in Iran [12,13]. Nonetheless, it seems that members of specific genera such as *Orchis*, *Anacamptis*, and *Ophrys* in addition to *Dactylorhiza*, *Himantoglossum*, *Neotinea*, *Platanthera*, and *Serapias* are usually common between different regions [8]. In agreement with the latter observation, nine members of the genera *Dactylorhiza*, *Anacamptis*, *Orchis*, and *Himantoglossum* were also reported in the present investigation (Appendix A). 

Previous studies from Greece reported that at least 14 orchid species from 4 genera were used for salep in Greece in older times, further highlighting that up to 7 species from 3 genera were recorded in modern times [14]. With wild sourcing of salep tubers in Greece dating back to the 1800s [22], and still practiced to date [14,17], a change in the utilization of *Dactylorhiza* species in modern times has been suggested, at least in Greece [14]. The latter assumption is based on increased rarity after the overharvesting of more appreciated orchids such as the members of genera *Anacamptis* and *Ophrys*, which were considered much more abundant in Macedonia and Thessaly in older times [22]. Although perhaps true, no such decline can be found in modern studies due to the absence of monitoring programs for wild-growing orchid populations. *Dactylorhiza sambucina* is considered the most commonly over-collected salep orchid in Greece to date [17]. According to interviewees from northern Greece reported in the literature [17], the over-collection of *D. sambucina* by the local community has decreased but harvesting by people outside the local community has increased. However, this is coupled with obscure differences in orchid abundance compared with the past despite possibly variable population sizes or diverse population densities at different times [17].

Along this line and with the aim to protect orchid resources, previously published recommendations have suggested the need for long-term monitoring of wild-growing salep orchid populations in Greece and elsewhere and effective modeling regarding the response of different salep species to different harvesting pressures [17]. Other recommendations include the implementation of collection bans in over-collected areas; strengthening and monitoring current regulations and how they are enforced; the development of a straight-forward DNA barcoding-based molecular identification system to monitor and track species diversity in trade combined with controls at customs offices [13]; the establishment of specific important orchid conservation areas (in parallel with IPAs—Important Plant Areas, see https://www.plantlifeipa.org/about, accessed on 1 May 2023) [41] for in situ conservation with the monitoring of wild-growing populations; the training of local stakeholders on sustainable collection practices; and the establishment of small-scale pilot cultivation to alleviate the effect of over-harvesting [13]. With only a few of these recommendations extant in Greece to date, we developed ongoing cooperation with Greek companies interested in sustainably cultivated salep orchids of Greece, providing expertise on sustainable conservation collections as well as ex situ propagation and cultivation of the selected species reported herein in artificial environments. Conservation-wise, the Natura 2000 sites in northwestern Greece currently offer in situ protection for the wild-growing salep orchid populations. The natural conditions of the Natura 2000 network in several areas of Greece and elsewhere may create suitable circumstances for the existence of many orchid taxa [42], and many of these orchids are included in the appendices of the EU Directive 92/43/EEC as “Other Important Species”, thus facilitating suitable management actions and ensuring possibilities for the survival of these plants in the future.

## 3. Materials and Methods

### 3.1. Authorized Collections of Wild-Growing Samples

Plant samples of Greek orchids used for salep were collected from mountainous areas in northwestern Greece (mainly) during 2018–2022 (Figure 4, Figure 5 and Appendix A). 

Information regarding the localities and habitats where salep orchids naturally grow in the wild was obtained through informal conversations with (a) local residents of mountain areas in northwestern Greece who collect bulbs for personal use and (b) shepherds of the Sarakatsani ethnic Greek subpopulation group who traditionally graze their flocks at high altitudes and are aware of the local flora [14]. These people traditionally collect bulbs and consume salepi, practically discerning the species in relation to the quality of the drink they produce. Local shepherds accompanied the research team to difficult-to-access localities (Figure 5c,d), and sometimes horses were used to transport the collected plant material. 

All collections were performed between May and early July each year. When not possible to collect living plant individuals, dried inflorescences were collected (Figure 5e,f, Appendix A). The wild-growing Greek orchid samples were in situ photographed to assist taxonomic identification and were collected after obtaining special permission (permits 154553/1861 and 182336/879 of 13-7-2017, 182336/879 of 16-5-2019, and 64886/2959 of 6-7-2020) issued by the national competent authority, namely, the Greek Ministry of Environment and Energy. After collection and taxonomic identification, an IPEN (International Plant Exchange Network) number was assigned to each sample maintained in the ex situ collections (salep individuals, Figure 5e,f and Appendix A) at the Balkan Botanic Garden of Kroussia, Institute of Plant Breeding and Genetic Resources, Hellenic Agricultural Organization—Demeter (ELGO-DIMITRA).

### 3.2. Plant Nomenclature

To allow comparisons across geographical scales and with other studies, all the species and subspecies are hereby cited with their currently accepted names according to the POWO (Plants of the World Online) database (https://powo.science.kew.org, accessed on 1 May 2023) [43] and not according to the Vascular Plants Checklist of Greece (https://portal.cybertaxonomy.org/flora-greece/intro, accessed on 1 May 2023) [39]. The latter online source was consulted for the local distribution of the studied orchids in the phytogeographical regions of Greece.

### 3.3. DNA Extraction, Amplification of Barcoding Regions, and Sequencing

Total genomic DNA was extracted using a modified CTAB protocol [44]. The extraction method was modified by facilitating cell lysis, extending the preparation time, and increasing the number of steps in DNA purification. Alterations and optimizations were performed regarding the added concentration of PVP-40, RNase, proteinase, and β-mercaptoethanol. Fresh tissue used for grinding under liquid nitrogen included leaves, flowers, tissue culture, and young plantlets, while dried tissue included inflorescences, whole plants, and whole tubers (Appendix A).

Four barcoding regions were tested, namely, *Internal Transcribed Spacer* (*ITS*) [45], *trnH-psbA intergenic spacer region* [46], *maturase k (matK)* [47] and *ribulose-bisphosphate carboxylase gene* (*rbcL)* [48]. The used PCR primers are reported in Appendix A. PCR reactions were prepared as follows: 1× KAPA Taq buffer (KAPA BIOSYSTEMS), 0.2 Mm Dntp mix, 0.4 Μμ of each primer, 1U/μL of KAPA Taq DNA Polymerase (KAPA BIOSYSTEMS), 20 to 40 ng DNA, and water to a final volume of 50 μL. The reaction conditions were 3 min at 95 °C, and 35 cycles of 30 s at 95 °C, 30 s at 53 °C, and 1.15 min at 72 °C followed by a final extension step of 1 min at 72 °C. All PCR amplicons were run in a standard 1.5% agarose gel where they were checked for specificity and quantity using a standard DNA ladder (markers). The amplicons were then cleaned up either from the gel or directly using a kit (Nucleospin Gel and PCR-Clean-up, Macherey-Nagel) and were sent for Sanger sequencing using a commercial provider. The same primers used for the PCR reactions were used for sequencing.

### 3.4. Analysis of Data

Sequencing data were manually checked for their quality, and contigs were assembled using Bioedit software [40] under default parameters. All generated sequences were used in Basic Local Alignment Search Tool (BLAST) [49] searches of the NCBI database. BLAST is regularly used for detecting sequence similarity in DNA barcoding projects. The identification of genera was performed using the top BLAST hit identifications provided by at least two DNA barcoding markers. The identification of species was performed when molecular data coincided and were corroborated with taxonomic identification. The further bioinformatic analysis included multiple sequence alignments (using CLUSTALW) and phylogenetic analysis using the UPGMA method [50]. The bootstrap consensus tree inferred from 500 replicates [51] was considered to represent the phylogeny of the taxa analyzed [51]. Branches corresponding to partitions reproduced in less than 50% of bootstrap replicates were collapsed. The percentage of replicate trees in which the associated taxa clustered together in the bootstrap test (500 replicates) was shown next to the branches [51]. The phylogenetic distances were computed using the maximum composite likelihood method [52] and were in the units of the number of base substitutions per site. This analysis involved 22 nucleotide sequences. All ambiguous positions were removed for each sequence pair (pairwise deletion option). There was a total of 1821 positions in the final dataset. The phylogenetic analyses were conducted in MEGA11 [45]. Sequences from *Diuris sulphurea* R.Br., *Goodyera schlechtendaliana* Rchb. f., and *Caladenia reticulata* Fitzg. were used herein as outgroups based on previous references from the literature (Figure 3).

## 4. Conclusions

The investigation herein has shown that DNA barcoding markers widely used in different orchid-related studies can also be used for the identification of wild orchids collected for salep by the Sarakatsani ethnic Greek population subgroup in northwestern Greece. The combined molecular barcoding (*ITS* and *matK*) and taxonomic classification of orchids collected from different sites in northern Greece showed that 53% represent members of the genus *Dactylorhiza*, 33.3% are members of the genus *Anacamptis,* and 11.8% are members of the genus *Orchis*. Considering the reasons that sometimes create boundaries in DNA barcoding, reliable projects require holistic approaches given that biological and technical complications/issues are inevitably insurmountable. The use of such straightforward DNA barcoding protocols for salep orchids may assist in tracking, monitoring, and regulating the trade of wild-harvested products (fair and traceable salep trade) and can also facilitate the conservation of natural populations of Greek salep orchids. 

Although in situ conservation of these orchid plants is ensured in various sites of the Natura 2000 network in Greece, there are still many wild-growing salep orchid populations outside the protected zones that suffer extensive over-collection directly from the wild. Therefore, long-term monitoring of wild-growing salep orchid populations is needed in Greece, which should be combined with ecological modeling regarding the response of different salep species to varied harvesting pressures. In addition, collection bans should be enforced in severely affected areas and in cases of illegal collections and exports from the country, while close monitoring of species diversity in extant salep trade should be coupled with informed controls at domestic customs offices. Training on sustainable collection practices and facilitation for the establishment of small-scale pilot cultivations of salep orchids are already in place in Greece, and these should be extended to foreign stakeholders with the aim to alleviate future collection pressure on wild-growing populations of these protected phytogenetic resources. 

## Figures and Tables

**Figure 1 plants-12-03038-f001:**
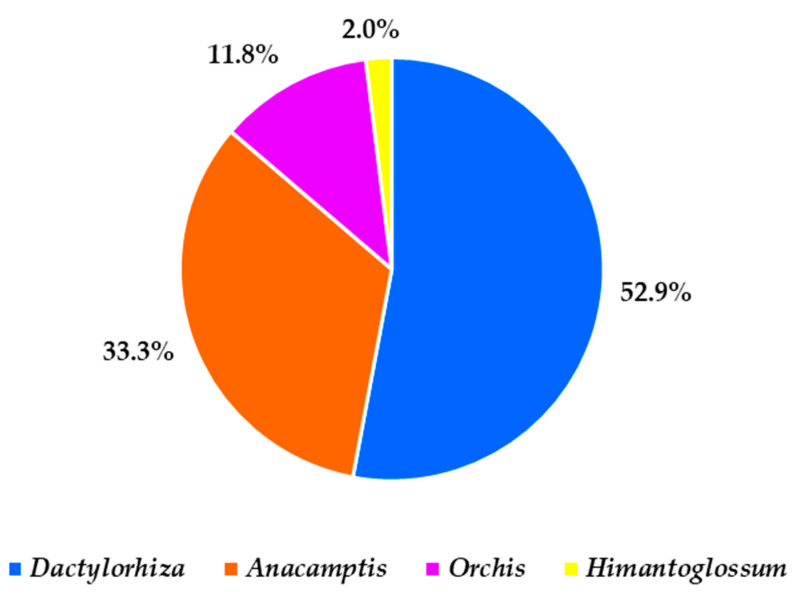
The proportion of different genera of Orchidaceae identified in wild-growing salep samples from northwestern Greece.

**Figure 2 plants-12-03038-f002:**
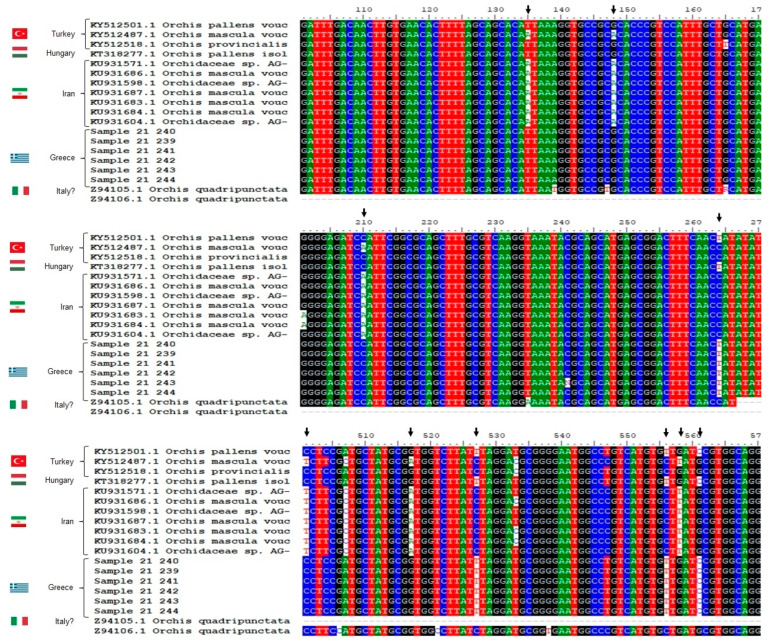
Multiple sequence alignment (CLUSTALW) of the *ITS* regions of the orchid samples from northwestern Greece (samples GR-1-BBGK-21,239 to GR-1-BBGK-21,244; only the last four or five digits are shown in the graphs) compared with NCBI-retrieved sequences belonging to different species of the genus *Orchis* (*Orchis* spp.). The country where the samples were collected is reported beside each of the sequences (Italian entries with no indication of the collection site in the database). Arrows indicate nucleotides that differ significantly among sequences, for example, in position 556, where the sequences of the Greek samples and *O. pallens* have thymine (T), while the sequences of *O. mascula*/*O. provincialis*/*O. quadripunctata* have cytosine (C). Alignment was edited using Bioedit software [40].

**Figure 3 plants-12-03038-f003:**
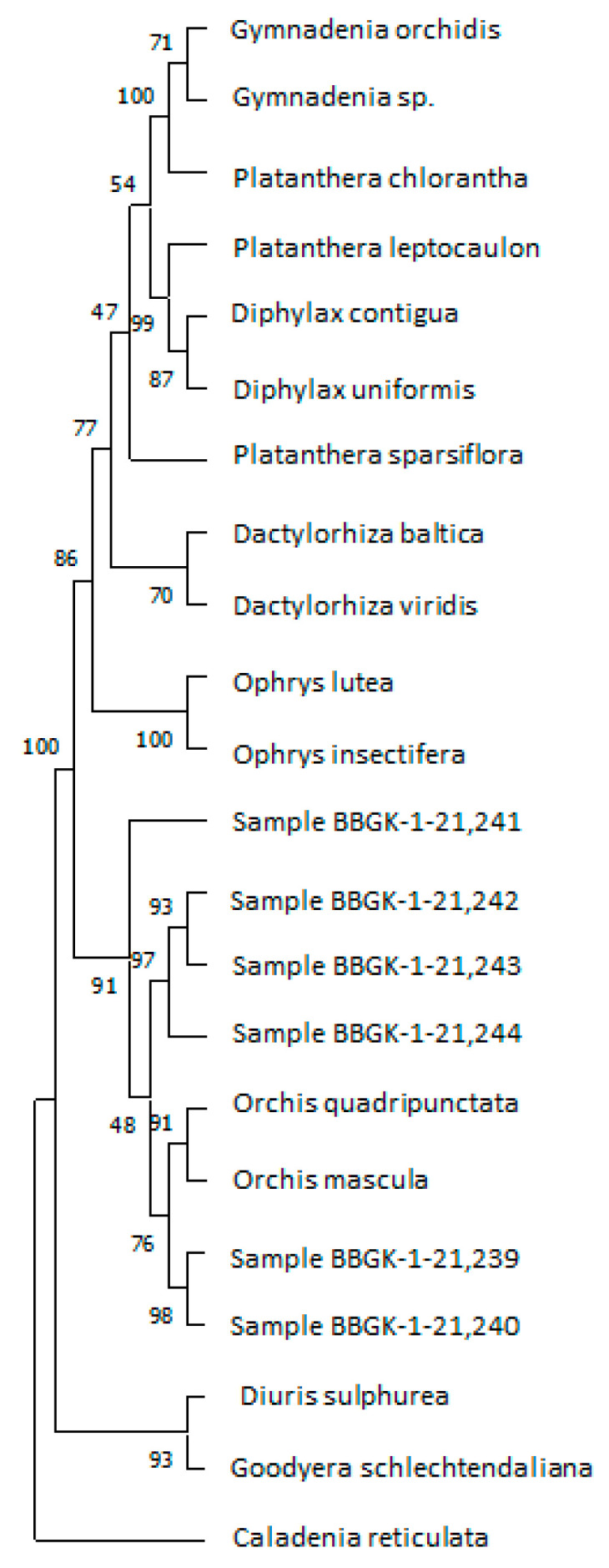
Phylogenetic comparison using the *matK* marker for the Greek samples of *Orchis* spp. and NCBI-retrieved data for members of other genera in Orchidaceae (Appendix A).

**Figure 4 plants-12-03038-f004:**
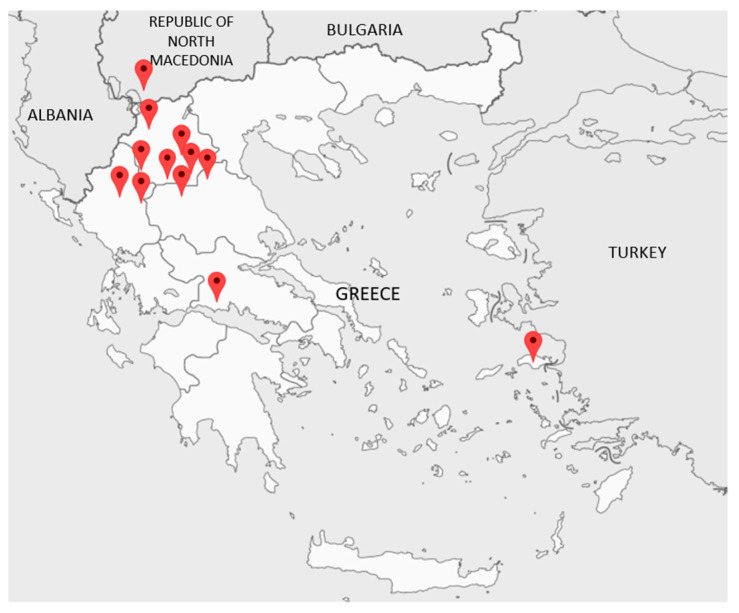
Collection areas of the different Orchidaceae specimens used for salep making with an emphasis on northwestern Greece (see also Appendix A).

**Figure 5 plants-12-03038-f005:**
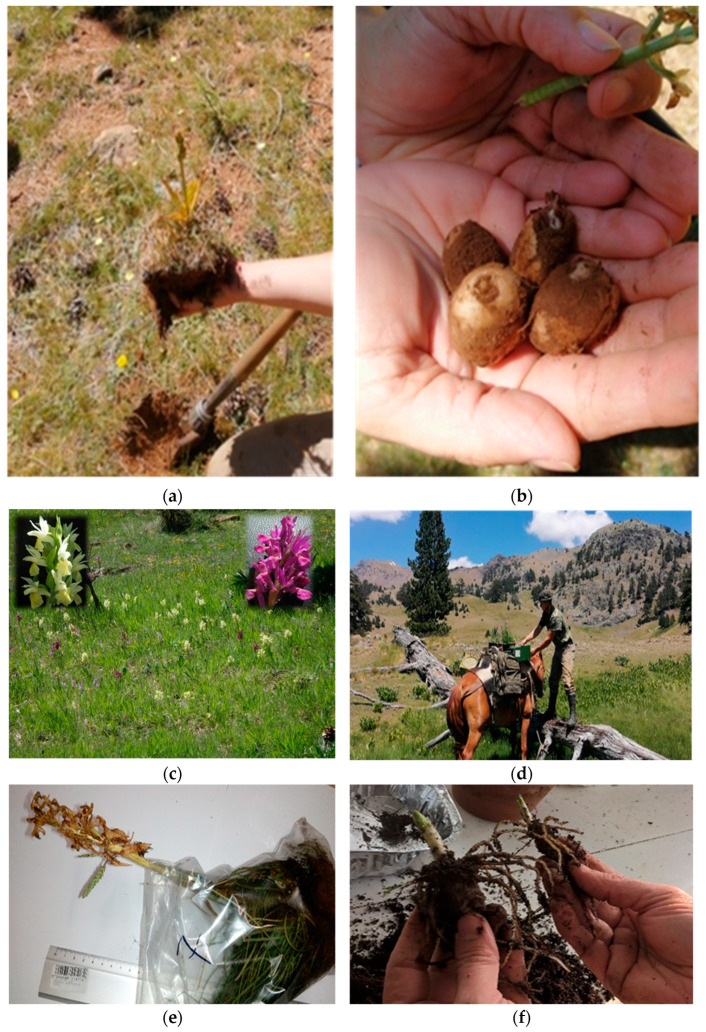
Collection, transport, and handling of wild-growing Greek orchid samples for ex situ conservation at the premises of the Institute of Plant Breeding and Genetic Resources, Agricultural Organization Demeter (Thermi, metropolitan Thessaloniki, Greece): (**a**,**b**) Collection of wild-grown *Orchis* sp. tubers from Mt. Smolikas, northwestern Greece. (**c**,**d**) Wild-growing *Dactylorhiza sambucina* individuals, with typical yellow flowers bearing light reddish stains (**left inset**) and purple flowers speckled with darker spots on the labellum (**right inset**), and transport of collected living orchid samples in difficult-to-access areas of Mt Smolikas, northern Greece. (**e**) Plant individual collected from the wild. (**f**) Separation of individual orchid tubers.

**Table 1 plants-12-03038-t001:** Grouping of different identification cases of the studied wild-grown salep samples (Orchidaceae) from northwestern Greece (*n* = 53) with consensus levels and respective reasons for the taxonomic and/or molecular approaches performed in this investigation (see Appendix A).

Consensus and Explanation	Cases	Reason
Consensus at the species level (Agreement between taxonomic and molecular identification)	3	Full taxonomic identification and undoubtful matching in molecular identification
Consensus at the genus level (Agreement between taxonomic and molecular identification)	7	Out-flowered specimens; matched only at the genus level
Partial consensus (Molecular identification aided with taxonomic identification)	29	Further identification of subspecies; verification, rejection, or resolution of possible matches
Partial consensus (Taxonomic identification aided with molecular identification)	9	Out-flowered specimens
Only taxonomic identification (no molecular identification)	2	Destroyed DNA samples
Only molecular identification (no taxonomic identification)	3	Only dry specimens

## Data Availability

Data are available on request due to restrictions. The data presented in this study are available on request from the corresponding author. The data are not publicly available due to privacy restrictions.

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
