# Peer review of "Taxonomic Identification and Molecular DNA Barcoding of Collected Wild-Growing Orchids Used Traditionally for Salep Production"

_plants, 2023, doi:10.3390/plants12173038_

Round 1

Reviewer 1 Report

The paper is interesting and sufficiently described, the study is well-prepared and concerns endangered and valuable plant species. I have only a few minor remarks, as follows.

1. Lines 49-51
The Authors wrote: "Although this preparation originated from the Eastern Mediterranean and the Balkans, it returned to be famous across Europe during the Renaissance period following the trend after the publication of Gerard’s Herbal in 1633 [7]."

Could be the salep produced using other terrestrial orchids? Do you know about its production in other regions of Europe? In general, terrestrial orchids are endangered and many species are going extinct.

2. Orchid symbionts
Orchids form a specific type of mycorrhizal symbiosis (orchid mycorrhiza) with fungi. Did you think about the study on orchid mycorrhizal symbionts and their potential role in the survival of tested (and potentially endangered) orchid taxa?

3. Line 171
Please, put this table into Supplementary materials, and instead, please add the map with localities. Maps are more readable and moreover, the distance between stands can be noticed. I understand the precise data on the distribution of studied orchids are sensitive data, but a map with marked regions would still be better.

4. Lines 261-463
The Authors wrote:  "establishment of specific important orchid conservation areas (in parallel with IPAs – Important Plant Areas, see  https://www.plantlifeipa.org/about) [44] for in situ conservation with monitoring of wild growing populations"

Are these recommendations appropriate for other EU countries? Some species like Dactylorhiza sambucina, occur in other EU countries and could be collected and imported in order to reduce the negative impact of orchid harvesting on local populations of orchids and their natural habitats in Greece.

None

Author Response

Journal: Plants (MDPI)

Manuscript Number: plants-2483093

Manuscript Title: Taxonomic identification and molecular DNA barcoding of collected wild orchid species used for salep traditional products

Response to Reviewer #1 comments

All the authors are thankful to the Editor’s and reviewers’ constructive comments that helped us improve the quality of our manuscript. In the revised manuscript, all changes, and new amendments are highlighted with green color throughout the manuscript.

Please find our response to your comments below:

Reviewer 1

The paper is interesting and sufficiently described, the study is well-prepared and concerns endangered and valuable plant species. I have only a few minor remarks, as follows.

1. Lines 49-51 The Authors wrote: "Although this preparation originated from the Eastern Mediterranean and the Balkans, it returned to be famous across Europe during the Renaissance period following the trend after the publication of Gerard’s Herbal in 1633 [7]."Could be the salep produced using other terrestrial orchids? Do you know about its production in other regions of Europe? In general, terrestrial orchids are endangered and many species are going extinct.

Authors’ answer: The most frequently harvested species for salep in Greece are Anacamptis morioDactylorhiza sambucina, D. saccifera and Orchis mascula, thus salep can be produced using other terrestrial orchids as well. In recent times, orchid harvesting for salep became popular again. Excessive collecting in Turkey and other Anatolian countries has raised serious conservation issues for wild orchid populations, while in Greece the rising market demand for natural organic products has increased the demand for salep. It must be said that 1000-4000 tubers are needed for a kilo of salep flour, depending on the species. A single cup of salep needs about 13 orchid tubers. Yet, the process of orchid harvesting for salep and its trade hides an irony; all wild orchids are protected by international and national legislation (e.g., CITES Convention, European Directive 92/43/EC, etc.), that strictly prohibits orchid collection, translocation, cut and further process of the material (Charitonidou et al. 2019). For all the aforementioned reasons, we conducted this study for the further cultivation of terrestrial orchids used for salep production.

  1. Orchid symbionts: Orchids form a specific type of mycorrhizal symbiosis (orchid mycorrhiza) with fungi. Did you think about the study on orchid mycorrhizal symbionts and their potential role in the survival of tested (and potentially endangered) orchid taxa?

Authors’ answer: This was not the case in this study, we attempted to determine the identity of Greek wild-growing orchid species based on molecular DNA barcoding combined with botanical taxonomy as employed tools for ex situ conservation of collected Greek orchids used for salep production as well as for the regulation of fair salep trade. However, another study is under progress investigating the existence of symbiotic fungi and bacteria in orchid tubers and soil-microbiota in their natural environment from which collected as well as we currently investigate the effect of different mycorrhizal symbionts formulations for the survival and further vegetative development and tuber - root growth of in vitro plantlets (derived from asymbiotic seed germination in vitro) to ex vitro greenhouse conditions and later on in open-field soil conditions.

  1. Line 171 Please, put this table into Supplementary materials, and instead, please add the map with localities. Maps are more readable and moreover, the distance between stands can be noticed. I understand the precise data on the distribution of studied orchids are sensitive data, but a map with marked regions would still be better.

Authors’ answer: As suggested by the reviewer, Table 1 has been transferred to supplementary materials as Table S1 and a map with the collection areas in northwestern Greece and surrounding countries was included in the revised manuscript as Figure 5.

  1. Lines 261-463 The Authors wrote:  “establishment of specific important orchid conservation areas (in parallel with IPAs – Important Plant Areas, see  https://www.plantlifeipa.org/about) [44] for in situ conservation with monitoring of wild growing populations”. Are these recommendations appropriate for other EU countries? Some species like Dactylorhiza sambucina, occur in other EU countries and could be collected and imported in order to reduce the negative impact of orchid harvesting on local populations of orchids and their natural habitats in Greece.

Authors’ answer: Many orchid species (i.e., Dactylorhiza sambucina) are collected and imported in Europe from Turkey, Iran and the Balkans, however the wild-growing local populations in these countries could be declined after long-term uncontrolled harvesting and salep trade, thus these orchid species could be at risk and endangered with extinction. See for more information also in reference list [i.e. 7, 10, 15, 19-22] of the revised manuscript. The aim of this study was the identification of different Greek Orchidaceae species and afterwards the evaluation of the proper cultivation conditions for each taxon (i.e., propagation, cultivation, testing mycorrhizal symbiosis and linked to survival and vegetative development/ tuber-root growth attributes, ex situ conservation, among others).

Reviewer 2 Report

The manuscript entitled “What is wild salep? Taxonomic identification and molecular DNA barcoding of collected wild orchid species used for salep traditional products” presents a study to assign the targeted orchids collected in specific genera or species through the combination of molecular DNA barcoding and botanical taxonomic identification.

The work is properly organized and structured, with a clear understanding of the results, easy to follow and logically explained, with proper conclusions (avoiding any speculation) based on the data obtained.

To my understanding, the methodology applied is adequate and complete. In conclusion, the results presented by the authors are of scientific relevance, with interesting conclusions so, I recommend its publication in Plants.

Minor editing of English language required

Author Response

Journal: Plants (MDPI)

Manuscript Number: plants-2483093

Manuscript Title: Taxonomic identification and molecular DNA barcoding of collected wild orchid species used for salep traditional products

Response to Reviewer #2 comments

Reviewer 2

The manuscript entitled “What is wild salep? Taxonomic identification and molecular DNA barcoding of collected wild orchid species used for salep traditional products” presents a study to assign the targeted orchids collected in specific genera or species through the combination of molecular DNA barcoding and botanical taxonomic identification.

The work is properly organized and structured, with a clear understanding of the results, easy to follow and logically explained, with proper conclusions (avoiding any speculation) based on the data obtained.

To my understanding, the methodology applied is adequate and complete. In conclusion, the results presented by the authors are of scientific relevance, with interesting conclusions so, I recommend its publication in Plants.

Authors’ answer: All the authors are grateful to receive reviews that help us reinforce the quality of our manuscript. Following your previous comments as well as the specific notes of the Academic Editor and the recommendations of the other reviewer, we corrected the text adopting most of the comments suggested. In the revised manuscript, all changes, and new additions in text and throughout the manuscript are highlighted with green color.
